# Source-Free Domain Adaptation Using Neighborhood Signature–Based Prediction Matching

## Abstract

Source-Free Domain Adaptation (SFDA) is an emerging area of research that aims to adapt a model trained on a labeled source domain to an unlabeled target domain without accessing the source data. Most of the successful methods in this area rely on the concept of neighborhood consistency but are prone to errors due to misleading neighborhood information. In this paper, we explore this approach from the point of view of learning more informative clusters and mitigating the effect of noisy neighbors using a concept called neighborhood signature, and demonstrate that adaptation can be achieved using just a single loss term tailored to optimize the similarity and dissimilarity of predictions of samples in the target domain. In particular, our proposed method outperforms existing methods in the challenging VisDA dataset while also yielding competitive results on other benchmark datasets.

## 1 Introduction

Deep learning models trained on labeled data are known to perform very well on a wide range of tasks. However, one key assumption for most models is that the test data comes from the same distribution as the training data (Valiant, 1984; Vapnik, 1995). These distributions are also referred to as "domains," as data collected from a particular domain tends to follow some specific distribution. When the test domain is different from the training domain, the performance of the models deteriorate, often drastically, due to the distribution shift (Gretton et al., 2009; Quionero-Candela et al., 2009). As a result, there has been a growing body of research in the area of Unsupervised Domain Adaptation (UDA) which seeks to adapt a model trained on a source domain to a target domain using unlabeled data from the target domain, which are generally easier to collect than labeled data (Ganin & Lempitsky, 2015; Gong et al., 2012; Bousmalis et al., 2017; Ganin et al., 2016). However, most UDA methods also require the labeled data from the source domain during the adaptation (Daumé III, 2007; Long et al., 2018; Shu et al., 2018). This can be impractical in many scenarios where we only have access to the source model and not the source data due to privacy or copyright reasons. To tackle this, a more restrictive setting has been explored, namely, Source-Free Domain Adaptation (SFDA) where we only have access to the trained source model and the unlabeled target data.

In this paper, we specifically focus on SFDA in the context of image classification. Several methods have thus far been explored for SFDA in image classification (Kundu et al., 2020a; Li et al., 2020a; Liang et al., 2020; Sun et al., 2020; Chen et al., 2018; Yang et al., 2021a; 2022). A defining characteristic of many of these methods is the reliance on neighborhood consistency in the feature space (Liang et al., 2020; Yang et al., 2021a; Litrico et al., 2023). The assumption here is that despite the domain shift, the source model forms clusters based on the semantic similarity of the samples from the target domain in the feature space (Yang et al., 2021a) and this can be used to encourage similar predictions for samples that are neighbors of one another in the feature space. Existing methods often leverage this through techniques such as estimating pseudo labels by aggregating the predictions of neighbors (Liang et al., 2020; Litrico et al., 2023) or directly enforcing neighborhood consistency with multiple neighbors (Yang et al., 2021a; 2022; Hwang et al., 2024) along with additional objectives such as diversity (Liang et al., 2020) and contrastive loss (Litrico et al., 2023; Wang et al., 2024).

The key limitation of these methods is that they are prone to learning misleading labels due to noisy neighbors, as there is no guarantee that most neighboring samples of a sample are correctly classified at any point in the training. In this paper, we propose a new method for this task to mitigate the effect of noisy neighbors. Our key idea is to use neighborhood consistency in a more reliable manner. To this end, we propose some key insights that can be used to estimate the pairwise semantic similarity and dissimilarity of samples in a training batch in the target domain, which ultimately leads to the model estimating the classes of these samples.

Our core approach is to align the predictions of the samples in a batch with respect to cosine similarity such that the samples that are estimated to be more semantically similar have greater cosine similarity in their predictions and vice versa instead of directly mapping samples into classes. This encourages the model to consider a more holistic view of the semantic information and enables more flexibility in dealing with difficult samples that have overlapping semantic characteristics with multiple classes. In order to estimate the semantic similarity, we introduce a scheme called neighborhood signature, which in our case is simply the mean prediction of the neighborhood samples, but unlike previous methods, we don't use the neighbors to directly estimate training pseudo labels. Another idea we propose is to encourage the model to learn more informative clusters, i.e., learn semantic differences of samples even when they are estimated to belong to the same class. The goal is to prevent the model from taking shortcuts and assigning samples to classes based on the most common semantic characteristics of the class and thus avoid confidently fitting samples to the wrong class due to noisy neighbors. For example, a model may learn to classify any yellow vehicle as a bus while ignoring the possibility of it being a car or a truck. The model has a better chance of avoiding this if it recognizes the variations within the bus, car, and truck classes. This can also implicitly prevent mode collapse without a separate diversity objective. The third consideration in our method is to consider the confidence with which samples are predicted and use this as inertia while aligning the predictions. For example, if we believe that two objects are semantically similar, and we are confident that one is a cat, and we are less confident that the other is a dog, then it is more likely that both are cats rather than both are dogs. Finally, we also consider the effect of class imbalance. Since different classes may have different frequencies of occurrence while aligning the predictions, the classes with greater frequency can have a dominating effect on the other classes; hence, we counter this by scaling the effect of samples based on their estimated class and the estimated frequency of the classes, even though these estimated classes are not directly used as training labels. Despite all of the above factors, our method still boils down to a single loss term where all the above factors are encoded in a mask that dictates the strength with which the pairwise cosine similarities of the predictions of samples should increase or decrease. Experimental results on benchmark data show the efficacy of our method, notably outperforming existing methods in the challenging VisDA dataset.

In Summary, our contributions are:

- We propose a novel approach to the SFDA problem that involves aligning cosine similarities of sample predictions based on neighborhood consistency, intra-class diversity, confidence, and class imbalance, all encoded into a single loss term.

- We conduct experiments on benchmark data and demonstrate the effectiveness of our method.

## 2 Related Work

**Domain Adaptation** Early Domain Adaptation methods mostly relied on the idea of aligning the feature distributions of the source and target domains. CORAL(Sun et al., 2016) uses moment matching. DANN Ganin et al. (2016) uses adversarial training to fool a domain discriminator while maintaining discriminativeness of features. CDAN Long et al. (2018) uses conditioning on discriminative information to assist adversarial feature alignment. DIRT-T Shu et al. (2018) uses adversarial training with the added constraint that samples belonging to clusters in the target domain belong to the same class. Additionally, Lee et al. (2019); Lu et al. (2020); Saito et al. (2018) are based on the idea of employing multiple classifier heads to give correct outputs in source domain and diverse outputs in the target domain and then aligning the domains through adversarial training. More recently, self supervision based methods have gained more traction. These methods often leverage pseudo labels generated by the model in the target domain. One early example is CST Liu

et al. (2021) which improves pseudo labels through a reverse training strategy based on making the pseudo labels more useful for the source model.

**Source-free Domain Adaptation** In Source-free Domain Adaptation, the source training data is unavailable during the adaptation, and so we must rely only on the source model. Most well known methods in this setting rely on self supervision. Early methods such as USFDA Kundu et al. (2020a) and FS Kundu et al. (2020b) leverage synthetic samples. SHOT Liang et al. (2020) learns pseudo labels based on nearest class centroids combined with entropy minimization and diversity maximization. 3C-GAN Li et al. (2020a) collaborates between a conditional generator and a classifier to gradually adapt the classifier to target styled samples. $A^2$-Net Chen et al. (2018) employs separation of source similar and source dissimilar samples and applies category wise matching. NRC Yang et al. (2021a) is a pseudo label based method that matches predictions with nearest neighbours while prioritizing reciprocal neighbours. AAD Yang et al. (2022) introduces the the idea of maximizing prediction mismatch with background samples. SF(DA)$^2$ Hwang et al. (2024) applies data augmentation in the latent feature space based on augmentation graphs and learns high quality clusters. SiLAN Wang et al. (2024) is another latent augmentation based method that uses contrastive learning and guides the augmentation using the feature dispersion statistics of the source model. While these methods employ various techniques to incorporate additional information in the training and improve cluster reliability, none can fully eliminate the problem of noisy neighbours. Our method explores an alternative approach which can move this field forward by improving the utilization of neighbourhood information.

**Deep Clustering** Deep Clustering refers to clustering very high dimensional data by first reducing them to a low dimensional feature space using a deep neural network. Earlier methods include DAC Chang et al. (2017) which does self supervised clustering of images based on pairwise similarity estimates which are in turn adapted by training a deep neural network. DCCM Wu et al. (2019) also uses pairwise similarity while improving the category estimates using pseudo labels along with maximizing local transformation robustness and mutual information. DEC Xie et al. (2016) is an auto-encoder based deep clustering method which assigns soft clusters based on cluster centroids and optimizes the KL-divergence between the clusters and an auxiliary target distribution. CC Li et al. (2020b) applies contrastive learning at both instance level and cluster level to directly map instances to cluster vectors.

## 3 Method

For source-free domain adaptation (SFDA), we are given a source-pretrained model and an unlabelled dataset with $N_t$ samples from the target domain as $D_t = \{\boldsymbol{x_i^t}\}_{i=0}^{N_t}$. Both the source domain and the target domain have the same $C$ classes. The goal is to adapt the model to the target domain. We denote the model as consisting of two parts: the feature extractor $f$, and the classifier $g$. The output of the feature extractor is denoted as the feature $(\boldsymbol{z_i} = f(\boldsymbol{x_i}) \in \mathbb{R}^h)$, where h is the dimension of the feature space. The output of the classifier is denoted as $(\boldsymbol{p_i} = \delta(g(\boldsymbol{z_i})) \in \mathbb{R}^C)$ where $\delta$ is the soft max function. We denote $P \in \mathbb{R}^{bs \times C}$ as the prediction matrix in a mini-batch, where $bs$ is the batch size. In order to do the adaptation, we formulate a loss function based on the cosine similarity of the prediction of each sample and the class encoding of all other samples in the mini batch. The loss function can be represented as:

$$L = \sum_{\substack{0 \leq i < bs \\ 0 \leq j < bs \\ i \neq j}} \boldsymbol{p_i} \boldsymbol{q_j} M_{ij} \tag{1}$$

where $\boldsymbol{p_i}$ is the model prediction of the $i^{\text{th}}$ sample and $\boldsymbol{q_j}$ is the class encoding (discussed later) of the $j^{\text{th}}$ sample and $M_{ij}$ is the mask that determines the magnitude and direction of the rate at which $\boldsymbol{p_i}$ and $\boldsymbol{q_j}$ should align.

We now describe the class encoding $\boldsymbol{q_j}$ and the various components of the mask $M_{ij}$

**Neighborhood Signature** As mentioned previously, we are leveraging the concept of neighbourhood consistency, i.e. we expect the model initialized as the source model to extract meaningful features in the target domain which causes semantically similar samples to map close to one another in the feature space. How-

ever, instead of directly estimating training labels from the potentially noisy neighbourhood, we work with the assumption that semantically similar samples have similar neighbourhood. The intuitive basis for this assumption is that, if the model confuses a tree beside a road with a lamp post beside a road then the same model is likely to confuse a tree beside a house with a lamp post beside a house and hence both tree samples would end up with similar looking lamp post samples in their neighbourhood even if the tree samples are not close to one another because one is influenced by road features and the other is influenced by house features (as the model is not already adept in extracting only relevant features in the target domain). Thus, we can gain meaningful information about the semantics of a sample by observing its neighbours. We refer to this information as neighbourhood signature. The question now is, how to encode the neighbourhood information to achieve this signature ? This can be explored in details in future work, but for the purpose of this paper we use the most obvious encoding, which is taking the mean prediction of the neighbourhood samples. Thus, for any given feature $z_i$, its neighbourhood signature is given by:

$$s_i = \frac{\sum_{k \in N(i)} p_k}{|N(i)|} \tag{2}$$

where $N(i)$ is the set of neighbours of the $i^{\text{th}}$ sample and $p_k$ is the model prediction for the $k^{\text{th}}$ sample as stored in the score bank. Note that, taking mean prediction of neighbours is a common approach, but our method differs in the motivation for it and the way it is used.

**Confidence-Adaptive Class Encoding** The class encoding for a sample is a tensor that encodes the estimated semantic information in a sample in terms of the probability of it belonging to each of the classes. In this sense it is similar to the pseudo label which is used in domain adaptation methods but in our case we do not try to fit samples to their own class encoding, rather we align the predictions of samples with the class encoding of other samples in a mini batch based on their estimated semantic similarity. One choice for class encoding is to use the neighbourhood signature itself since in our case it of the same dimension as the output probabilities, but this risks reinforcing noisy neighbourhoods, thus leading to misleading semantic estimates. To resolve this, we use a simple solution which is to choose either the current prediction of the sample or the neighbourhood signature, depending on which ever has the lesser prediction entropy. Therefore, the class encoding $q_j$ of the $j^{\text{th}}$ sample is given by:

$$q_j = \arg \min_{x \in \{s_j, p_j\}} H(x) \tag{3}$$

**Neighborhood Similarity** Now, in order to construct the mask $M_{ij}$, we use the hypothesis that the cosine similarity between the prediction of the $i^{\text{th}}$ sample $p_i$ and a class encoding $q_j$ in the loss should increase if samples $i$ and $j$ have similar neighbourhood and vice versa. The reasoning, as described before, is that semantically similar samples should have similar neighbourhoods. Therefore, one proposed component of the mask is:

$$M_{ij1} = 1 - 2 \cdot s_i \cdot s_j \tag{4}$$

As we can observe, for very similar neighbourhood ($s_i \cdot s_j$ approaching 1), this component approaches -1 which will strongly align $p_i$ and $q_j$ as we minimize the loss while for very dissimilar neighbourhood ($s_i \cdot s_j$ approaching 0) this component will approach 1 thus causing $p_i$ to move away from $q_j$.

Since we do not yet know how to perfectly encode the neighbourhood information so that our mask in Eq 1. equals the ideal mask, we improve upon the raw masks achieved by our mean based neighbourhood similarity with the following additional terms based on practical observations. These terms are not exhaustive, and the core idea provides the scope to add more terms and engineer the mask for better adaptation.

**Intra-Class Diversity** As mentioned earlier, we want to encourage intra-class diversity to encourage the model to learn semantic variations within a class. For this, we add a second component to the mask $M_{ij}$ as:

$$M_{ij2} = \alpha \cdot s_i \cdot s_j \tag{5}$$

Here, $\alpha$ is a decay term that is initialized as 1 and decays to 0 as the training progresses. This term forces samples with similar neighbourhood to be unaligned and the combined effect of this and the previous term

leads to a weak repulsive force between samples which are semantically similar and likely belong to the same class with this repulsion being stronger for more dissimilar samples. This term is decayed out into order to prevent it from hindering the formation of compact clusters towards the end of the training.

**Inertia of Class Encoding** We propose that when a prediction is being aligned with respect to the class encoding of another sample (whether aligning towards or away), the strength of this alignment should depend on the confidence of the class encoding. Consider two samples $i$ and $j$ which are being estimated to be similar based on their neighbourhood signatures. If they have different class encoding, then it is more likely that the one with the more confident class encoding represents the correct label for both hence the prediction for the one with the less confident class encoding should be moved with greater strength towards the class encoding of the other sample. The same is true while moving predictions away from a class encoding if they are estimated to be dissimilar based on their neighbourhood signature. Thus, we scale the mask $M_{ij}$ with the confidence of the class encoding of the $j^{\text{th}}$ sample. Following Litrico et al. (2023), we represent the confidence of the $j^{\text{th}}$ sample as:

$$\gamma_j = \exp\left(-\frac{H(\boldsymbol{q_j})}{\log_2 C}\right) \tag{6}$$

However, we found empirically that doing this in the initial part of the training does not lead to improvements probably because the confidence values the less reliable during the early stages of training so we use this term in conjunction with the decay term $\alpha$. Therefore, the actual confidence score of the $j^{\text{th}}$ sample is:

$$\gamma_j' = \alpha + (1 - \alpha) \cdot \gamma_j \tag{7}$$

**Scaling for Class Imbharsharaj.pathak.ete@gmail.comalance** Since the classes may be imbalanced, the class encoding favouring the classes with more representation can dictate the alignment process. For example, if there are many samples in the car class but very few samples in the bus class, then bus samples will experience more frequent attractions towards car samples due to certain semantic similarities between the two compared to other bus samples which can cause bus samples to be misclassified as cars. To counter this, we scale the intensity of alignment towards samples on a batch-wise basis. For each mini batch, we make an estimate of the number of samples belonging to each class based on the *argmax* of the class encoding and then scale the mask $M_{ij}$ with the class frequency-based scaling factor of the $j^{\text{th}}$ sample which is given by:

$$w_j = \frac{1}{\alpha + (1 - \alpha) \cdot C_{\text{argmax}(\boldsymbol{q_j})} \cdot \frac{C}{B}} \tag{8}$$

Where $C_k$ is the number of samples in the current batch estimated to belong to class $k$. $C$ is the total number of classes and $B$ is the batch size. This means that samples belonging to smaller classes will on average, impart a stronger alignment force, and this effect will come into play later in the training process due to the introduction of $\alpha$

By putting together all of the above terms, the final loss function is given by:

$$L = \sum_{\substack{0 \le i < bs \\ 0 \le j < bs \\ i \ne j}} \boldsymbol{p_i q_j}(M_{ij1} + M_{ij2}) \cdot \gamma_j' \cdot w_j \tag{9}$$

Our algorithm is illustrated in Algorithm 1.

## 4    Experiments

**Datasets**. We conduct experiments on three benchmark datasets for image classification: PACS(Li et al., 2017), Office-31(Saenko et al., 2010) and VisDA-C 2017(Peng et al., 2017). **PACS** contains 4 domains (Art-Painting, Cartoon, Photo and Sketch) and 7 object categories with and 9991 images and a large domain shifts due to different styles. **Office-31** contains 3 domains (Amazon, Webcam, DSLR) with 31 classes and

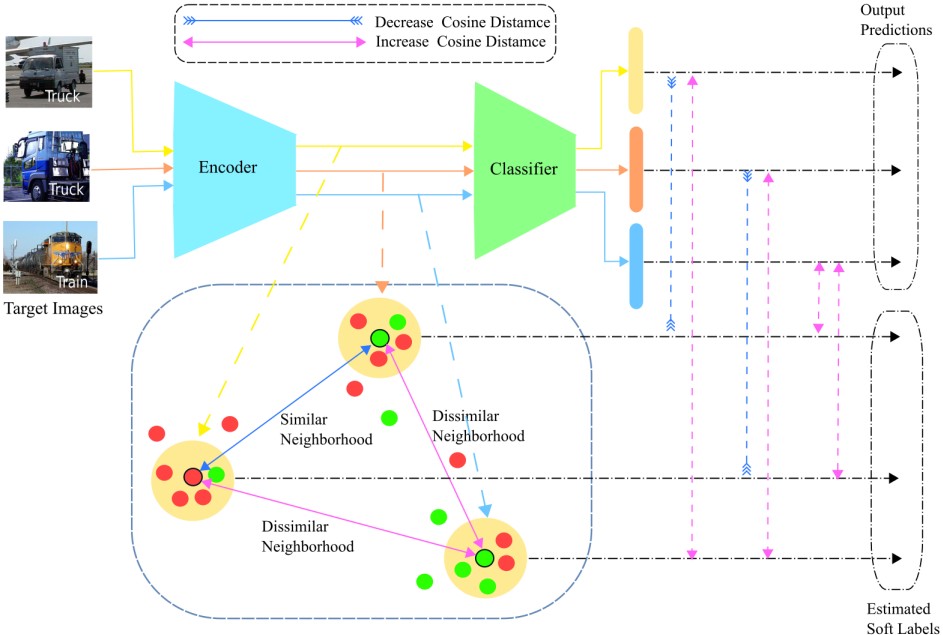

Figure 1: An overview of the proposed Source-Free Domain Adaptation (SFDA) approach. For a batch of data, the encoder maps the input images to the feature space, where we compute the neighbourhood signatures of the samples. Using these neighbourhood signatures, we estimate the class encoding of the samples, and then we optimize the cosine distance between the output prediction of each sample and the class encoding of all other samples in the batch as per our proposed loss function.

---

**Algorithm 1** Neighbourhood Signature Contrastive Adaptation

---

**Require:** Source-pretrained model and target data $D_t$
 1: Build memory bank storing all target features and predictions
 2: **while** Adaptation **do**
 3:     Sample batch $T$ from $D_t$ and update memory bank
 4:     **for** each sample $\boldsymbol{x_i}$ in $T$ **do**
 5:         Retrieve $K$-nearest neighbours ($N_i$) of $\boldsymbol{x_i}$ and their predictions from memory bank
 6:         Compute the neighbourhood signature $\boldsymbol{s_i}$ of $\boldsymbol{x_i}$ using Eq. 2
 7:         Compute the confidence-adaptive class encoding $\boldsymbol{q_i}$ of $\boldsymbol{x_i}$ using Eq. 3
 8:         Compute the inertia of estimated label $\gamma_j^{'}$ of $\boldsymbol{x_i}$ using Eq. 7
 9:         Compute the class frequency based scaling factor $w_i$ of $\boldsymbol{x_i}$ using Eq. 8
10:     **end for**
11:     Update model by minimizing Eq. 9
12: **end while**

---

4,652 images. **VisDA** is a challenging large scale dataset, having 12 classes with a source domain consisting of 152k synthetic images and a target domain consisting of 55k real object images.

**Experiment Setup.** To ensure fair comparison with previous methods, we use the same network architecture, training techniques and hyperparameters for all competing methods. For each experiment, the network consists of a feature extractor followed by a classifier. Specifically, we adopt the backbone(He et al., 2016) of a ResNet-18 for PACS, ResNet-50 for Office-Home and ResNet-101 for VisDA. We adopt SGD with momentum 0.9 and batch size of 64 for all datasets. The learning rate is set to 1e-3 for Office-31 and PACS and 1e-5 for VisDA. We train for 50 epochs for PACS, 40 epochs for Office-31 and 15 epochs for VisDA. There are two hyperparameters, the number of nearest neighbours $K$ and the decay rate $\alpha$. In all our experiments, we set $K$ to 5 and for the decay rate we use: $\alpha = (\frac{1}{2})^{\frac{\text{current iter}}{\text{num iter per epoch}}}$.

## 4.1 Results

We compare our method with existing SFDA methods on the above-mentioned datasets. For a fair comparison, we reproduce AaD, SF(DA)$^2$, and SiLAN using their official codes. As shown in Table 1, our method outperforms all others methods on VisDA in terms of average accuracy across the classes. Table 2 shows that our method gives comparable performance to other methods on Office-31 and Table 3 shows that our methods gives significantly better performance compared to other methods on PACS.

Table 1: Comparison of the SFDA methods using ResNet-101 on VisDA2017.

| Method | plane | bcycl | bus | car | horse | knife | mcycl | person | plant | sktbrd | train | truck | Avg. |
|---|---|---|---|---|---|---|---|---|---|---|---|---|---|
| SHOT(Liang et al., 2020) | 94.3 | 88.5 | 80.1 | 57.3 | 93.1 | 94.9 | 80.7 | 80.3 | 91.5 | 89.1 | 86.3 | 58.2 | 82.9 |
| HCL(Huang et al., 2021) | 93.3 | 85.4 | 80.7 | 68.5 | 91.0 | 88.1 | 86.0 | 78.6 | 86.6 | 88.8 | 80.0 | **74.7** | 83.5 |
| G-SFDA(Yang et al., 2021b) | 96.1 | 88.3 | 85.5 | 74.1 | **97.1** | 95.4 | 89.5 | 79.4 | 95.4 | 92.9 | 89.1 | 42.6 | 85.4 |
| NRC(Yang et al., 2021a) | 96.8 | 91.3 | 82.4 | 62.4 | 96.2 | 95.9 | 86.1 | 80.6 | 94.8 | 94.1 | 90.4 | 59.7 | 85.9 |
| AaD(Yang et al., 2022) | 95.2 | 90.5 | 85.5 | 79.2 | 96.4 | **96.2** | 88.8 | 80.4 | 93.9 | 91.8 | 91.1 | 55.9 | 87.1 |
| DaC(Chang et al., 2017) | 96.6 | 86.8 | 86.4 | 78.4 | 96.4 | **96.2** | **93.6** | **83.8** | 96.8 | **95.1** | 89.6 | 50.0 | 87.3 |
| SF(DA)$^2$(*Hwang et al.*, 2024) | 97.2 | 90.9 | 85.0 | 54.0 | 96.7 | 95.9 | 90.1 | 80.9 | 95.0 | 92.7 | 89.2 | 61.4 | 85.8 |
| AaD+SiLAN(Wang et al., 2024) | 96.4 | 90.4 | 86.1 | 80.0 | 96.9 | 92.3 | 86.8 | 83.0 | 94.7 | 91.7 | 88.5 | 46.1 | 86.1 |
| Ours | **97.6** | **91.7** | **86.7** | **84.8** | 96.7 | 94.9 | 91.5 | 81.5 | **97.5** | 93.6 | **92.5** | 47.0 | **88.0** |

Table 2: Comparison of SFDA methods using ResNet-50 on Office-31.

| Method | A→D | A→W | D→W | D→A | W→D | W→A | Avg. |
|---|---|---|---|---|---|---|---|
| SHOT(Liang et al., 2020) | 94.0 | 90.1 | 98.4 | 74.7 | 99.9 | 74.3 | 88.6 |
| NRC(Yang et al., 2021a) | 96.0 | 90.8 | **99.0** | 75.3 | **100.0** | 75.0 | 89.4 |
| 3C-GAN(Li et al., 2020a) | 92.7 | **93.7** | 98.5 | 75.3 | 99.8 | 77.8 | 89.6 |
| HCL(Huang et al., 2021) | 94.7 | 92.5 | 98.2 | **75.9** | **100.0** | 77.7 | **89.8** |
| AaD(Yang et al., 2022) | 95.0 | 92.1 | 99.0 | 75.7 | 99.8 | 76.3 | 89.6 |
| SF(DA)$^2$(*Hwang et al.*, 2024) | 94.3 | 88.5 | 80.1 | 57.3 | 93.1 | **94.9** | 82.9 |
| AaD+SiLAN(Wang et al., 2024) | 94.3 | 88.5 | 80.1 | 57.3 | 93.1 | **94.9** | 82.9 |
| Ours | **96.4** | 93.0 | 98.9 | 75.4 | 99.8 | 73.4 | 89.5 |

## 4.2 Analysis

**Ablation Study** To see the effectiveness of the various components of our method, we conduct ablation studies by removing individual components from the pipeline. The studies are conducted with respect to performance on the PACS dataset. Table 4 shows that each component is crucial in the achieving the best accuracy.

Table 3: Comparison of SFDA methods using ResNet-18 on PACS.

| Method | P→A | P→C | P→S | A→P | A→C | A→S | Avg. |
|---|---|---|---|---|---|---|---|
| NEL(Ahmed et al., 2022) | 82.6 | 80.5 | 32.3 | **98.4** | 84.3 | 56.1 | 72.4 |
| AaD(Yang et al., 2022) | 84.8 | 76.0 | 57.2 | 90.0 | 82.4 | 63.2 | 75.6 |
| SF(DA)$^2$(*Hwang et al.*, 2024) | 87.3 | 82.7 | 54.1 | 91.3 | 83.1 | 60.3 | 76.5 |
| AaD+SiLAN(Wang et al., 2024) | 87.4 | **85.1** | 43.7 | 90.3 | 83.5 | 64.3 | 75.7 |
| Ours | **90.9** | 78.2 | **57.6** | 98.0 | **90.0** | **68.6** | **80.6**silan |

Table 4: Ablation studies of subcomponents of the proposed method measured by classification accuracy (%) on PACS.

| Method | Avg. |
|---|---|
| Ours | 80.6 |
| Ours - Intra-Class Diversity | 80.2 |
| Ours - Inertia of Estimated Labels | 79.7 |
| Ours - Class-Wise Scaling | 78.4 |
| Ours - Confidence-Adaptive Class Encoding | 66.3 |

**Effect of number of nearest neighbours** We observe the effect of different choices for the number of nearest neighbours $K$ for performance on the PACS dataset. Figure 2 shows that the performance is robust to various reasonable choices of $K$.

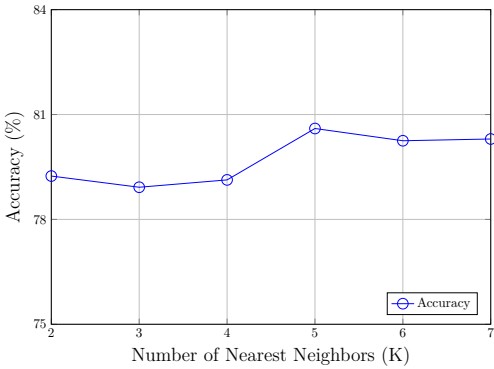

Figure 2: Accuracy vs the number of nearest neighbors (K) on PACS

**Effect of the decay base** In our proposed implementation, we use $\frac{1}{2}$ as the base of the decay factor $\alpha$. We observe the effect of different choices of this base with respect to performance on the PACS dataset. Figure 3 shows that the performance is robust to various reasonable choices for the base of $\alpha$.

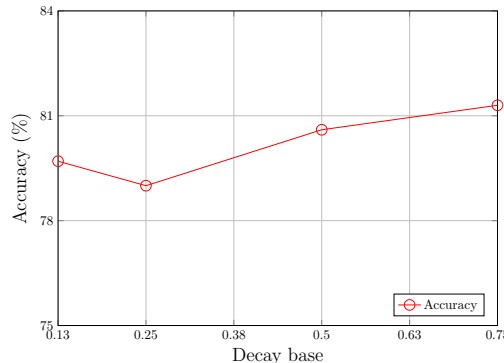

Figure 3: Accuracy vs the decay base for $\alpha$ on PACS

## 5 Conclusion

We proposed to tackle source-free domain adaptation using a novel approach based on aligning predictions of samples based on their neighbourhood signature. We further propose the idea of incorporating intra-class diversity along with prediction inertia and class weight based scaling to design a training objective with a single loss term. Furthermore, we show that the proposed method achieves state-of-the-art performance on several benchmark datasets and also conduct subsequent analysis on the method.

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
