# OpenReview forum: "Source-Free Domain Adaptation Using Neighborhood Signature–Based Prediction Matching"
_TMLR — Withdrawn by Authors_

### Review · Reviewer_3rj5 · 2025-08-11

**Summary Of Contributions:**

This paper tackes source-free domain adaptation: a regime in which a pre-trained model is adapted to new target data for which there are no labels, and the pre-training data is not available either. The paper proposes a neighborhood-based approach; each sample is encoded based on the model output on the sample as well as the average of model outputs on neighboring samples (with neighbors determined in an intermediate layer feature space). These two encodings are combined in a number of ways to produce a loss function that encourages samples with similar classes to align. This includes a neighborhood similarity mask by which samples with similar neighborhoods are encouraged to align, as well as a confidence-based term which boosts confident encodings over less confident ones. Experiments demonstate superior performance relative to baselines.

**Audience:**

Yes

**Audience Explanation:**

Given the relatively strong empirical findings, I believe the SFDA community would be interested in this paper. In particular, this paper introduces a number of innovations, some of which could inspire the development of new techniques in the field.

**Broader Impact Concerns:**

No broader impact concerns.

**Claims And Evidence:**

No

**Claims Explanation:**

Given that the paper is purely empirical, the quality of the empirical results is very important. Unfortunately, I believe there are a few deficiences that make the claims in the paper less than convincing:

First, standard errors/deviations are not reported for the tables, and error bars are not reported for the figures. This makes it difficult to assess the statistical significance of the results.

Second, the results in Table 2 seem to show that the proposed method is not the best: a number of methods seem to perform better in the average column.

Finally, while the authors do provide a number of ablations in Table 4, I believe these are not sufficient given the number of components in the proposed method. In particular: for ours - confidence-adaptive class encoding, I would also like to see variations where 1) the class encoding is always given by the model prediction, 2) the class encoding is always given by the neighborhood signature. Also, there does not appear to be an ablation where neighborhood similarity is removed. A third ablation that is important is determining the neighborhood by the model output instead of in the feature space. Overall, more ablations are needed to justify the complexity of the proposed approach.

**Requested Changes:**

**Critical**
- Standard deviations/errors and error bars for tables and figures
- Additional ablations (see above)


**Would strengthen**
- There are a few typos in the paper that need to be fixed (for example in Tables 1, 2, after equation 7 etc.)
- Please place Figure 1 earlier in the paper; this will help with building intuition for the reader

---

### Review · Reviewer_6GtU · 2025-08-25

**Summary Of Contributions:**

The paper proposes a source‑free domain adaptation (SFDA) method that aligns the cosine similarity of predictions between sample pairs using a single, unified loss. The loss consists of a mask that encodes (i) neighborhood similarity via a **neighborhood signature** (mean prediction of K‑NN in a memory bank), (ii) **intra‑class diversity** to discourage collapse early in training, (iii) **confidence “inertia”** such that stronger class encodings guide weaker ones, and (iv) **class‑imbalance scaling** at the batch level. A confidence‑adaptive class encoding chooses, per sample, between its current prediction and its neighborhood signature by lower entropy, avoiding reinforcement of noisy neighborhoods. Empirically, the method seems to achieve strong/competitive results on 3 commonly used benchmark datasets for SFDA.

**Strengths**:

* S1: The proposed objective (Eq. 9) is simple and straightforward. Each of the four components (Intra-Class Diversity, Inertia of Estimated Labels, Class-Wise Scaling, Confidence-Adaptive Class Encoding) seem to improve performance on commonly used benchmark datasets for SFDA.

**Weaknesses**:

* W1: Eq. 1 includes $p_i q_j M_{ij}$ but never states explicitly whether $p_i q_j$ is a dot product or (normalized) cosine similarity. The text alternates between “cosine similarity” and “cosine distance,” leading to ambiguity about what is actually minimized and how normalization (if any) is handled for probability vectors.
* W2: The paper motivates the mask components intuitively, but provides no formal analysis. Claims like “implicitly prevent mode collapse” are not backed by clear evidence. Given the very small gain of "Intra-Class Diversity" (Table 4), the benefit of this needs to be shown in more detail.
* W3: The loss sums over all $(i,j)$ pairs in a batch (Eq. 1), which is $O(B^2)$. No discussion of computational/memory complexity of the proposed method.
* W4: The ablation (Table 4) removes components individually on the PACS dataset only.
* W5: Standard deviations of the results are not included. Did you average over many seeds? This is particularly important since the average gains are over competing methods are relatively small.
* W6: On VisDA, the accuracy on e.g. the “truck” class is low (47.0) compared to several baselines >50-61. The reason for this is not analyzed.
* W7: In **Table 2 (Office‑31)** the rows for *SF(DA)²* and *AaD+SiLAN* are **identical**, which is unlikely correct.
* W8: The setup says ResNet‑50 for *Office‑Home* even though the dataset section lists *Office‑31* as used. Clarify which dataset is actually evaluated.
* W9: Some baseline methods are not included for some datasets. Why is this?

**Audience:**

Yes

**Audience Explanation:**

The proposed loss (Eq. 9) includes some interesting ideas (that are novel to the best of my knowledge), and the results indicate some empirical benefit.

**Broader Impact Concerns:**

None.

**Claims And Evidence:**

No

**Claims Explanation:**

See list of weaknesses regarding limitations of the evidence/experiments. Given the listed weaknesses, I do believe the paper need to improve the experiments (which they have plenty of space to include: only 8.5 out of 12 pages are used).

**Requested Changes:**

Refer to weaknesses (W1-W9) above. All weaknesses are critical, except W8.

---

### Review · Reviewer_Ro3L · 2025-08-27

**Summary Of Contributions:**

This paper presents a new method for Source-Free Domain Adaptation designed to overcome issues with "noisy neighbors" in existing approaches. The core idea is to use a "neighborhood signature" (the average prediction of a sample's neighbors) to estimate semantic similarity.

The method employs a single loss function that aligns sample predictions based on four key factors: neighborhood similarity, intra-class diversity, prediction confidence, and class imbalance. This unified approach encourages the model to learn more robust and informative clusters in the target domain.

Experiments show the proposed method achieves state-of-the-art performance on the challenging VisDA and PACS benchmark datasets and performs competitively on Office-31.

**Audience:**

Yes

**Audience Explanation:**

I think it will attract some interest for two key reasons:

1. This paper tackles Source-Free Domain Adaptation, a relevant and practical problem in modern AI.
2. It makes the bold claim of achieving new state-of-the-art performance on challenging benchmarks, which obligates researchers in the field to read and evaluate it.

**Claims And Evidence:**

No

**Claims Explanation:**

- The reported baseline results in Table 1 seem problematic. They are substantially different from those in the original SiLAN [1] paper, where SiLAN alone (without AaD) achieves 88.3% on VisDA. The paper claims to reproduce baselines using their "official codes," which makes this discrepancy problematic. This issue may cast doubt on the central claim of achieving state-of-the-art performance, as their method may only be superior to a weakened or poorly implemented version of the actual baseline.


- The method attempts to correct for class imbalance by calculating class frequencies within each mini-batch (Equation 8). With a batch size of 64, this is a noisy and unstable estimate of the true global class distribution in the target domain. A single unrepresentative batch could dramatically skew the loss function, potentially harming stability. A more robust approach would use a running average of class predictions over many batches.

- The core assumption is that a neighborhood's average prediction (the signature) is more reliable than a single neighbor's prediction. While this provides some smoothing, it does not solve the fundamental problem of error propagation. If a model consistently misclassifies a group of visually similar samples (e.g., classifying all leopards as cheetahs), their neighborhood signatures will be nearly identical and incorrect, causing the model to reinforce its own mistake. The method is still vulnerable to this kind of confirmation bias, especially in the early stages of adaptation.

[1] What Has Been Overlooked in Contrastive Source-Free Domain Adaptation: Leveraging Source-Informed Latent Augmentation within Neighborhood Context

**Requested Changes:**

See above.

---

### Note · Authors · 2025-09-01

I have read and agree with the venue's withdrawal policy on behalf of myself and my co-authors.